# Lasers in Medicine: The Changing Role of Therapeutic Laser-Induced Retinal Damage—From *de rigeuer* to Nevermore

Jeffrey K. Luttrull 

Ventura County Retina Vitreous Medical Group, 3160 Telegraph Rd Suite 230, Ventura, CA 93003, USA; jkluttrull@proton.me; Tel.: +1-805-650-0664; Fax: +1-805-650-0865

**Abstract:** For over five decades, laser-induced retinal damage (LIRD) was thought to be the necessary cost of all therapeutic effects of laser treatment for the most important causes of irreversible visual loss, the chronic progressive retinopathies (CPRs). The development of modern retinal laser therapy with the discovery of "low-intensity/high-density subthreshold micropulse" laser (SDM) showed that the supposed need for LIRD represented a case of confusing association with causation. This revealed that LIRD was unnecessary and detrimental to clinical outcomes, and thus, contraindicated as the most severe complication of retinal laser treatment. SDM allowed for an understanding of the mechanism of retinal laser treatment as a physiologic reset effect, triggered by heat-shock protein (HSP) activation upregulating the unfolded protein response and restoring proteostasis by increasing protein repair by 35% in dysfunctional cells via a thermally sensitive conformational change in the $K_{10}$ step of HSP activation kinetics. Because HSP activation kinetics are catalytic, even low levels (the "reset" threshold) of HSP activation result in a maximal treatment response. SDM and the study of HSP activation kinetics in the retina show that the therapeutic effects of retinal laser treatment can be fully realized without any degree of LIRD. Besides LIRD, all effects of retinal laser treatment are restorative and therapeutic, without any known adverse treatment effects. Without LIRD, the benefits of retinal laser treatment are infinitely renewable and direct treatment of the fovea is possible. Elimination of LIRD from retinal laser treatment has revolutionized the clinical potential of retinal laser treatment to broaden treatment indications to permit, for the first time, effective early and preventive treatment to reduce visual loss from the most frequent causes of irreversible visual loss worldwide, the CPRs.

**Keywords:** laser; retina; photocoagulation; subthreshold; micropulse



## 1. Introduction

"More doctors smoke Camels than any other brand."
-Magazine advertisement from the 1950s
Things change.

A historical overview shows a trend of surgical cures being replaced by medical therapies. Antibiotics replace amputation, chemotherapy replaces excision. Such transitions reflect our increasingly better understanding of biology, chemistry, and pathophysiology. But rather than simply reflecting advances in science, these transitions also reflect the increasing value placed on human life, the desire to minimize human suffering, and the desire to maintain form and function. Until very recent times, pain, suffering, death, and disability were close companions of rich and poor alike and accepted as part of everyday life. Thus, surgery without anesthesia, and surgical maiming and mutilation, if seen as advantageous, were more easily accepted as necessary and unavoidable.

Despite this, change, even for the better, is usually resisted in favor of the status quo. Consider the resistance to surgeons' hand washing in the 19th century [1]. The cost? At least one U.S. president. Some types of change are easier to accept and more welcome than

others. "If you take these pills, we might not have to cut off your foot", for instance; others, less so.

Retinal laser treatment is a particularly interesting example of such dynamics. This paper examines the transition from the belief that laser-induced retinal damage (LIRD) was an absolute prerequisite to therapeutic effectiveness, to the current understanding that LIRD is an unnecessary complication of treatment, how this was determined, and the implications of this transition from surgery to medicine. However, at the very same time, retinal laser treatment is somewhat atypical, and possibly even unique. This atypia has made progress in the field more difficult, more easily resisted, and more poorly understood than in most other cases. This is because in addition to the typical resistance to change, there is confusion [2]. As will be discussed, even though retinal laser treatment has made the transition from surgery to medical therapy, it remains "retinal laser treatment". Seldom, if ever, has a therapeutic intervention in medicine transitioned from a clearly surgical procedure to an essentially medical therapy while maintaining precisely the same mechanism of action and name. Few confuse amputation with penicillin. But the vast majority of ophthalmologists remain confused about retinal laser treatment two decades after the debunking of retinal laser treatment as surgery, i.e., after demonstrating that LIRD was sufficient, but unnecessary, for effective retinal laser treatment [3,4].

The following will discuss the changing role of LIRD in retinal laser treatment, the therapeutic implications, and the vagaries of science along the way. Note that in this discussion, we do not consider retinal laser for cautery, such as to treat retinal tears or to destroy small tumors, but focus instead on treatment of the most common and important indications for retinal laser treatment, the neurodegenerations of the chronic progressive retinopathies (CPRs) [5,6].

## 2. Light for Cautery

Ancient civilizations used focused light to heat water, sink ships, and cauterize wounds. The ability of intense light to damage vision has thus been known since ancient times. In the 17th century, the first correlations with foveal burns were made in patients blinded by unprotected viewing of a solar eclipse. Beyond sun-gazing, the application of intense light for various purposes was achieved by use of glass lenses to focus sunlight [7]. Thus, long before ophthalmic application and the advent of the laser, the ability of high-intensity focused light to burn, cauterize, and destroy was well known.

These historical observations were confirmed by animal studies, such as those by Czerny (1867) and Deutschmann (1882) who were able to produce thermal retinal burns in rabbits using sunlight directed by mirrors and focused with lenses. Later in the 19th century, the advent of electrical arc lights resulted in frequent inadvertent retinal injuries. Maggiore used reflected sunlight to photocoagulate an ocular tumor in a patient anticipating enucleation, allowing for a post-enucleation histopathologic study that showed the now typical findings of thermal necrosis, inflammation, and hyperemia [8]. Gerd Meyer-Schwickerath was one of the first to explore and apply photocoagulation to retinal disorders in earnest, originally experimenting with sunlight, and then with the xenon arc, the first commercially available retinal photocoagulation device, made by Zeiss in the 1950s [9]. The publication of his work resulted in Meyer-Schwickerath being widely accepted as the pioneer of modern retinal photocoagulation ahead of others, such as Moron from Spain, who proceeded him but did not publish until later [10]. Using the xenon arc light source, Meyer-Schwickerath was able to treat retinal tears and holes, and was the first to recommend treatment of retinal vascular disease and vascular tumors, such as von Hipple–Lindau lesions. He was also the first to perform and recommend grid photocoagulation or "panretinal" photocoagulation for the treatment of diabetic retinopathy [9].

The development of the LASER (linearly amplified stimulated electromagnetic radiation) resulted from the work of a number of investigators, particularly Townes and Gould, leading to the Nobel Prize for physics in 1964 [11,12]. But it was Maiman's synthetic ruby crystal solid-state laser (wavelength 693 nm) in 1960 that introduced a clinically practical

laser for photocoagulation to the field of ophthalmology. By allowing better focusing, heating uniformity, and increased control compared to arc light sources, Maiman's ruby laser made ophthalmic application more feasible, and thus more useful and popular [13].

### 3. Laser-Induced Retinal Damage (LIRD)

What is apparent from the above is that the goal of intense light treatment of the retina, first using sunlight, then arc lights, and finally laser, was to burn (photocoagulate) the retina, either to destroy localized lesions, such as small tumors or retinal neovascular fronds, cauterize retinal breaks, or to debulk the retina, such as through panretinal application of the xenon arc in eyes with diabetic retinopathy. Light, and thence the laser, were seen as surgical tools, well suited for thermal destruction of intraocular structures because they could be safely introduced inside the eye in the clinic by projecting the light through the transparent anterior segment onto the retina. Thus, it was the desire and perceived need for LIRD that was the driving force in ophthalmic light, and finally laser, development [14].

### 4. Laser Retinal Photocoagulation (RPC)

Retinal photocoagulation (RPC) for complications of diabetic retinopathy (DR) and retinal vein occlusion (RVO) became major indications for retinal laser, principally as focal and grid RPC, while treatment of choroidal neovascular membranes (CNV), particularly associated with age-related macular degeneration (AMD), was the unique indication for suprathreshold ablative RPC in the macula [15,16] (Figure 1).

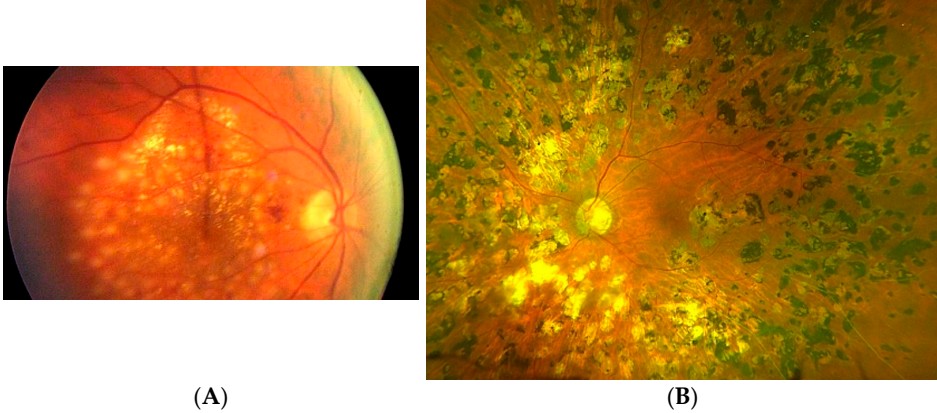

**(A)**                                      **(B)**

**Figure 1.** (**A**) 30-degree color ocular fundus photograph of acute suprathreshold macular retinal photocoagulation lesions. Note yellowish-white spots of full-thickness retinal thermal coagulation. An iatrogenic acute disseminated multifocal chorioretinitis. (**B**) 50-degree color ocular fundus photograph showing LIRD of multiple diffuse atrophic retinal laser lesions in eye following macular and peripheral retinal (panretinal) photocoagulation. These lesions expand gradually with time. Expansion of the LIRD lesions near the fovea may eventually lead to loss of central visual acuity. For scale, the horizontal diameter of the optic nerve is 1.5 mm. From: Luttrull JK. Modern Retinal Laser Therapy. Principles and Application. Kugler Publications, Amsterdam, The Netherlands August 2023 ISBN: 978-90-6299-298-0.

Despite steady progress in most other areas of ophthalmology, prototypical LIRD, i.e., photocoagulation, was believed to be the absolute prerequisite for therapeutic efficacy and remained so for over 50 years. This belief was universally accepted and questioned by no one of record during that time [17]. This is despite the fact that the causative necessity of LIRD for the therapeutic effects of retinal laser could not be established beyond association. That RPC "worked" in comparison to no treatment at all, and in light of the lack of viable alternatives for much of that time, disquiet over the difficulty of explaining the mechanism of action in which the premise of essential LIRD was rooted was tempered [18].

Despite the unanimity of opinion, there were clinical observations that cast doubt on the necessity of LIRD and its essential therapeutic role. These were clinical settings that

showed a therapeutic effect resulting from anything other than direct localized thermal retinal destruction. The most prominent were, consistent with the early observations of Meyer-Schwickerath, that grid photocoagulation of the retina could stop progression and achieve regression of diabetic retinopathy without direct treatment of areas of neovascularization [9,14,18,19]. Another was the observation that "indirect" photocoagulation, such as treating adjacent, to but not directly on, subretinal leaks in eyes with active central serous chorioretinopathy, could achieve disease resolution as effectively as "direct" treatment to the point of active leakage itself [20]. Both indicated effects beyond direct tissue destruction. Similarly, for diffuse diabetic macular edema, macular grid photocoagulation (indirect treatment) increased in popularity due to its effectiveness despite the frequent lack of specific targets for focal treatment and its ability to more effectively address diffuse edema [21]. Subsequently, the Early Treatment of Diabetic Retinopathy Study (ETDRS) confirmed that retinal laser complications and adverse treatment effects increased with increasing treatment intensity—generally suprathreshold photocoagulation employed in focal treatments including direct ablation of retinal neovascularization—while efficacy increased with increasing treatment density, and that grid treatment of diabetic macular edema (DME) was as effective as focal treatment (direct treatment) [22].

These realizations led to gradual reductions in treatment intensity and efforts to reduce neurosensory retinal damage with longer-wavelength krypton red and diode near-infrared lasers, and then microsecond-pulsed lasers (MPL) and short-pulse continuous wave (CW) lasers (SPL); they also led to an increase in the use of indirect grid treatments, leading finally to pattern scanning laser applications. Despite this, as described by Mainster, the goal of treatment remained steadfastly to cause LIRD thought to be the absolute requirement for a therapeutic effect [21,23–28] (Figure 2).

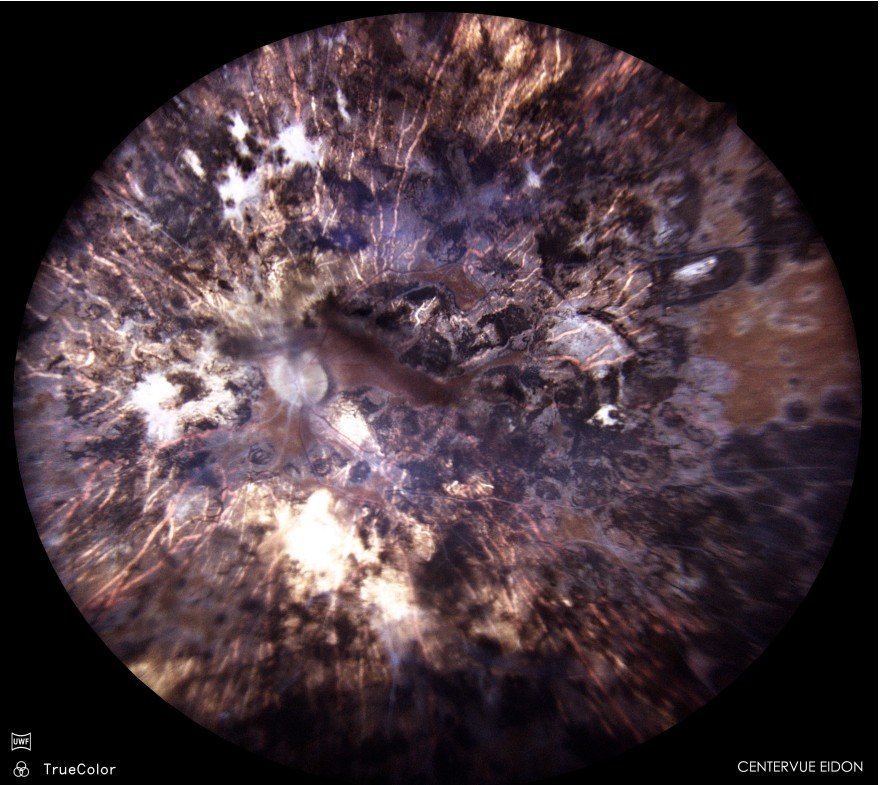

**Figure 2.** 50-degree color ocular fundus photograph of left eye of 75-year-old woman 50 years after near-total retinal photocoagulation for proliferative diabetic retinopathy. Visual acuity 20/80 with 2° central visual field. Note diffuse and confluent retinal laser damage. For scale, the horizontal diameter of the optic nerve is 1.5 mm. From: Luttrull JK. Modern Retinal Laser Therapy. Principles and Application. Kugler Publications, Amsterdam, The Netherlands August 2023 ISBN: 978-90-6299-298-0.

## 5. Traditional Theories for the Therapeutic Mechanism of LIRD

Three main theories, either alone or in combination, were proposed to explain the mechanism of retinal laser treatment, based on the presumptive prerequisite of LIRD: (1) It was suggested that by destroying full-thickness retina by laser thermal ablation with RPC, or parts of it such as the retinal photoreceptors, metabolic demand could be reduced, reducing physiologic stress on the retina, or diseased retina could be simply debulked. Less retina, less disease, was the thinking [27]. (2) Independently, and as a consequence of (1), it was proposed that thinning of the retina resulting from photocoagulation would facilitate increased oxygen diffusion from the choroid to the vitreous through the thinned, atrophic post-RPC retina, improving inner retinal oxygenation via diffusion through the vitreous. This theory was often invoked to explain the benefits of suprathreshold RPC for DR [29–31]. (3) Stimulation of biological factors, particularly such things as temperature-sensitive heat-shock proteins (HSPs), might lead to amelioration of the disease processes via a number of direct and indirect pathways [4,6,32]. Note that this last explanation for retinal laser effects is generic, potentially applying to any retinal laser mode and any disease state, not just retinal vascular disease which theories (1) and (2) emphasize. Also note that while (1) and (2) require retinal destruction, option (3) envisions laser-alteration of retinal biologic activity, requiring retinal preservation.

## 6. Thermal Effects on the Retinal Pigment Epithelium (RPE): The Common Denominator of All Retinal Laser Treatment

It is well established that the therapeutic effects of retinal laser are independent of wavelength. Near-infrared (NIR) wavelengths are not absorbed by the neurosensory retina. This means that thermal effects arising from laser absorption by RPE melanin, common to all retinal laser wavelengths, are the mediators of all therapeutic retinal laser effects [32–42]. Subsequent developments revealed that all therapeutic benefits of treatment could be produced without LIRD. These include reduction in vascular endothelial growth factor (VEGF) and increase in pigment epithelial derived factor [32–38]. By exclusion, this revealed that all therapeutic laser effects were mediated chemically by living RPE cells thermally affected, but not killed, by treatment. Further studies revealed thermal activation of RPE heat-shock proteins (HSPs) as the initiator of the therapeutic response. This exposed LIRD not as therapeutic, but as wholly detrimental, avoidable, and the most severe adverse effect of retinal laser treatment. The following examines how we arrived there, and how currently available clinical retinal laser platforms reflect this photobiological reality [32–38].

## 7. Current Laser Modes and Platforms for Retinal Laser Treatment and LIRD

### 7.1. Ultra-Short Pulse Lasers (USPLs)

USPLs were developed in the 1990s as part of the trend to reduce, but maintain, LIRD in an attempt to reduce adverse treatment effects compared to RPC, while maintaining efficacy (with LIRD still being considered essential at that time). The development of USPLs reflected the difficulty in adequately controlling responses to conventional RPC for this purpose (Figures 1 and 2). USPLs, such as the nanosecond-duration frequency-doubled "2RT" laser (Ellex, Adelaide, Australia), are essentially YAG lasers designed to photodisrupt the RPE [38]. Because of the ultra-short pulse, there is no time for heat dissipation or transfer from the melanosome, or for stimulation of biological effects such as heat-shock protein activation. Instead, the RPE cell is simply vaporized. The lack of time for heat transfer can spare adjacent structures such as Bruch's membrane and retinal photoreceptors. However, more violent RPE "explosions" can cause physical damage beyond the RPE, often evidenced by subretinal hemorrhages due to ruptures of Bruch's membrane [26,39]. Further, similar to photodisruption, like YAG laser photodisruption of the posterior capsule after cataract surgery, nanosecond lasers may cause a shock wave in the plane of the RPE which extends at least 1 mm in all directions from the laser spot. Such shock waves may account for drusen resorption at a distance from the nanosecond laser spots, and this trauma may contribute to the documented worsening of eyes with high-risk

dry AMD following USPL [26]. (Chang DB proprietary data Ojai Retinal Technologies, LLC, Los Angeles, CA, USA) A vaporized RPE cannot contribute to any therapeutic effect. Instead, the therapeutic effects of nanosecond laser arise from a wound-healing response and HSP activation in cells surrounding the denuded laser spot that are required to close the wound by sliding laterally to fill the defect. Remarkably, this has been described as "rejuvenation" and "non-damaging" (Ellex, Adelaide, Australia). Proponents of nano- (and microsecond) CW lasers point out that the healing response, and thus mechanism, of these laser modes is different than retina-sparing laser modes [40]. However, this difference is not a good one. Instead, it involves the activation of unique processes required to repair physical tissue damage, such as activation of tissue matrix metalloproteinases and their inhibitors that are required for remodeling of the laser wound [36,39–41]. Because of the absence of heat spread and the "cookie cutter"—like RPE defects created by nanosecond laser—the total area of affected but not killed retina able to contribute to the therapeutic effect is minimal, thus minimizing therapeutic benefits. Thus, nanosecond laser has very limited application, limited effectiveness, limited repeatability, and cannot be used in the fovea, all factors that limit clinical usefulness [42] (Figure 3) (Table 1).

**Table 1.** Comparison of characteristics of various clinical retinal laser modes.

| Laser Type | LIRD? | LIRD Mechanism | Primary Therapeutic Mechanism | Foveal Treatment? | High Density? | Repeatable? | SAEs? |
|---|---|---|---|---|---|---|---|
| 2RT | + | PD | W | - | - | L | + |
| SRT | + | PA | W | - | - | L | + |
| PASCAL | + | PA | W | - | - | L | + |
| CW RPC | + | PC | R | - | - | L | + |
| SDM | - | - | R | + | + | U | - |

CW = continuous wave. 2RT = nanosecond CW laser. SRT (selective retinal therapy) = microsecond pulsed laser. PASCAL (pattern scanning laser) = CW microsecond laser. SDM = low-intensity/high-density subthreshold diode microsecond pulsed laser. LIRD = laser induced retinal damage. SAE = serious adverse treatment effects. PD = photodisruption. PA = photoablation. RPC = retinal photocoagulation. W = wound healing/defect closure. R = reset effect. L = limited. U = unlimited.

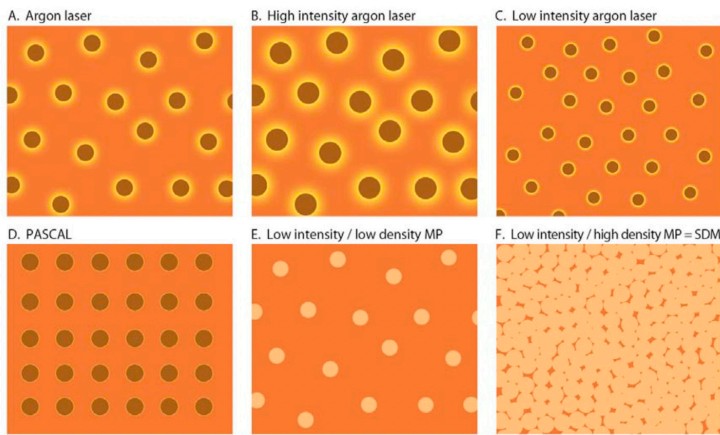

**Figure 3.** (**A–F**). Graphic representation of the "Effective Surface Area" of various modes of retinal laser treatment for retinal vascular disease. Vermillion = retina unaffected by laser treatment. Brown = area of retina destroyed by laser and inactive with respect to ability to produce extracellular cytokines. Yellow = area of retina affected by the laser but not destroyed, able to contribute to the therapeutic effects of laser treatment via laser-induced alteration/normalization of cytokine expression. PASCAL = pattern scanning laser. MP = diode micropulse laser. SDM = "High density/low-intensity" subthreshold/subvisible diode micropulsed laser. From: Luttrull JK, Dorin G. Subthreshold diode micropulse photocoagulation as invisible retinal phototherapy for diabetic macular edema. A review. Current Diabetes Reviews, 2012, 8, 274–284.

## 7.2. Short-Pulse Lasers

Short-pulse CW lasers (SPLs) were developed at the same time and for the same reasons as ultra-short-pulse lasers. Lengthening the pulse duration slightly to the microsecond range (3–12 μm) reduces the explosive effect of RPE melanosome heating, resulting instead in a thermomechanical steam-bubble formation at the melanosome and lethal internal cavitation of the RPE cell [37–45]. Like USPL, SPL generally employs a 532 nm wavelength. At sufficiently low power, rather that exploding the RPE cell, SPL kills the cell while preserving the cell membrane and limiting the damage to the RPE. Histopathologic studies in animals and humans reveal, however, that the RPE and outer retina are generally damaged as well. Healing of SPL lesions is similar to those from USPL, with sliding of adjacent RPE and outer retinal cells to fill the gaps created by SPL spots [45–48]. In the same way, the area of affected but not killed RPE at the margins of the SPL treated area is minimal, limiting clinical effectiveness, while the LIRD limits clinical usefulness, as with USPL (Figure 3). Photoacoustic monitoring of SRT treatment (Lutronix, Livermoore, CA, USA) has been developed to "hear" the laser-induced intracellular cavitations to help titrate exposures in order to guarantee cell destruction [47]. The pattern scanning laser (PASCAL, Topcon, Mountain View, CA, USA), originally developed with the goal of selectively destroying retinal photoreceptors, has implemented a titration algorithm now to attempt to avoid LIRD, called "Endpoint Management" (EpM) [39]. This uses subjective assessment of a "threshold" to test burning, from which the power is reduced to attempt to avoid LIRD with subsequent applications. Unfortunately, this is a theoretical possibility, but practical impossibility, for three reasons: First, the algorithm is based on subjective clinical assessment of photocoagulation-caused retina burn intensity, making the entire exercise unreliable and idiosyncratic. Second, the algorithm was developed based on rabbits, who have very uniform and homogeneous retinal pigmentation, unlike humans [37]. Third, the therapeutic window of CW lasers, like the PASCAL—the range between no effect and thermal retina damage—is only 0.010 watt in breadth—much too narrow and small a target to consistently hit clinically [32–35,37,39,40,44]. Thus, LIRD remains the rule with regard to microsecond lasers.

## 7.3. Conventional Retinal Photocoagulation

As noted above, RPC represents the earliest and remarkably durable traditional approach to retinal laser treatment, setting the standards and expectations for retinal laser treatment and all that followed. RPC uses CW lasers employing typically visible wavelengths, applied in millisecond duration exposures. As noted above, the therapeutic window for CW lasers is narrow—just 0.010 watt. For this reason, LIRD from RPC is the rule, as it is both the intent of RPC and nearly impossible to avoid, even if desired. When LIRD was thought to be the necessary precondition for therapeutic benefit, this made RPC ideal for treatment. The longer spot duration of RPC allows for heat transfer and dissipation into to the tissues surrounding the laser spot [23,31,35,37,38]. The degree of spread and collateral damage is directly related to the intensity and duration of the RPC spot application. Low-intensity applications may damage only the RPE, choriocapillaris, and outer retina ("subthreshold") [49–53]. Increasing the intensity extends the photocoagulative damage higher into the neurosensory retina ("threshold"). Thermal coagulation of the proteins in neurosensory retina transforms effected retina from clear to opaque due to light-scattering. Tissue temperatures used to achieve this effect are generally in the range of 50–70 °C [12,31]. Higher intensities will photocoagulate the full thickness of the retina, such as in the case of the treatment employed in the ETDRS ("suprathreshold") [22] (Figures 1–3). This degree of photocoagulation makes the laser burns appear clinically light grey or even white. Even higher intensity, especially if in shorter duration, can cause gas bubble formation and rupture Bruch's membrane, leading to a high risk of secondary choroidal neovascularization [50]. The depth and breadth of retinal damage caused by RPC means that healing of the defect is generally by fibrosis rather than sliding of adjacent cells such as the sliding that occurs in nanosecond and short-pulse microsecond CW exposures [49–53]. With increasing spot intensity, the "halo" of affected but not killed RPE adjacent to the RPC spot widens, increasing the therapeutic effects (Figures 1–3). Unfortunately, this is at the cost of more damage, functional loss, inflammation, and adverse effects that are proportional to the burn intensity and density. Thus, while the therapeutic benefits of RPC exceed nanosecond and microsecond laser, so do the adverse treatment affects, limiting applications and clinical usefulness and outcomes, particularly visual outcomes, which are especially sensitive to the loss of function and damage-induced inflammation caused by RPC [38,44,49–53] (Figure 4). These include reduced retinal and visual function such as decreased visual acuity, visual field loss, and impaired retinal function through electrophysiology, inflammation, macular edema, pain, burn-lesion creep, choroidal neovascularization, retinal and subretinal hemorrhage, sub and pre retinal fibrosis, and in severe cases uveitis, choroidal effusion, and serous retinal detachment [51].

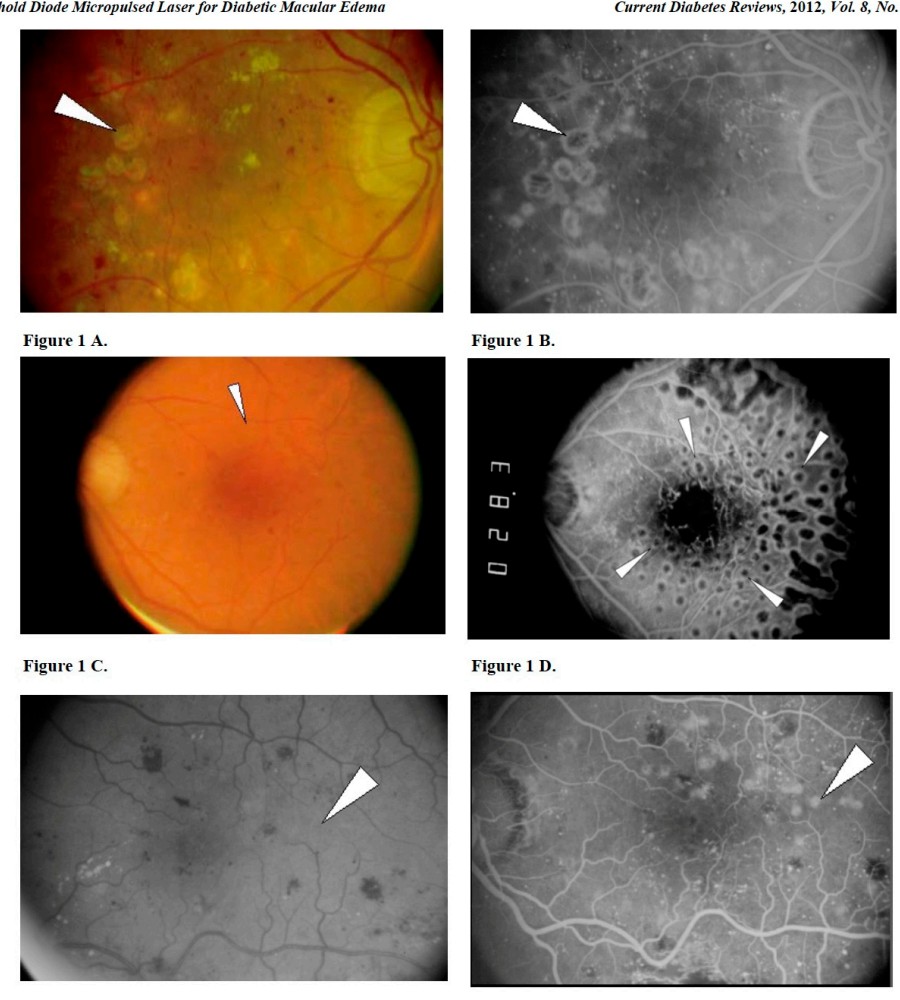

**Figure 4.** Illustrations of various degrees of retinal damage resulting from "subthreshold" retinal photocoagulation for diabetic macular edema. Fundus photographs are on the left, intravenous fundus fluorescein angiograms (FFA) on the right. Top row (A,B) show "subthreshold" laser damage (LIRD) easily visible clinically, on fundus photos, and by FFA. Rows C,D and E,F show "subthreshold" LIRD that is more difficult to see clinically and by fundus photography, but still easily seen by FFA. Arrows show FFA visible lesions that are variously visible (or invisible) by fundus photography. This figure is same as Figure 1 in Luttrull JK, Dorin G. Subthreshold diode micropulse photocoagulation as invisible retinal phototherapy for diabetic macular edema. A review. Current Diabetes Reviews, 2012, 8, 274–284.

### 7.4. Microsecond Pulsed Lasers (MPL)

Micropulsed lasers (MPL) can have a wide range of effects, from those associated with RPC to retina-sparing treatment sublethal to the RPE, depending on the mix of the various laser parameters, particularly wavelength, and pulse frequency ("duty cycle") [35,37,48]. Like UPSL and SPL, MPL—also introduced in the late 1990's—also attracted the attention of those seeking to reduce but maintain LIRD [23,24]. For MPL in general, the higher the duty cycle and shorter the wavelength, the more likely it is that thermal damage will occur [34,35,37,48]. For 810 nm, retinal damage at a 5% duty cycle or less in clinical power ranges and spot sizes is nearly impossible [35,37]. On the other hand, at duty cycles of 15% or more, damage is likely, as at these higher pulse frequencies it behaves clinically like a CW laser due to a marked shortening of the thermal relaxation time between microsecond pulses [37,48]. Higher energy wavelengths, like 577 μm and below, make retinal damage possible and even likely except at very low powers, increasing treatment risks and the potential for inadvertent retinal damage [23]. Because of a doubling of the RPE absorption and quadrupling of the energy of 810 nm, the therapeutic range of 577 nm is 8 times narrower than 810 nm for even a 5% duty cycle appropriately powered, thus markedly reducing the safety margin of treatment [35].

The length of the microsecond pulses in the pulse train of microsecond pulsed lasers is longer than short-pulse lasers, ranging 40–100 μs compared to just 3–12 μs for CW microsecond SPL lasers [38]. These longer

microsecond pulse durations allow time for heat dissipation and heat transfer from the RPE melanosomes within the cell, avoiding photodisruption and cavitation and thermomechanical cell death. The effects of microsecond pulsed lasers are entirely thermal [37–40]. Through appropriate spacing of the longer microsecond pulses, "off times" of sufficient duration between pulses can prevent excessive intracellular heating by allowing inter-pulse cooling [37]. Non-retinal damaging microsecond pulsed laser treatment generally increases average cell temperature a therapeutic 6 °C. Microsecond pulse pike temperatures of an additional 6–8 °C cause this average temperature-rise and are superimposed on top of it [35,48]. Because of the brevity of the spike temperatures, they remain sublethal to the cell. However, because of the acuity and severity of the microsecond pulse temperature increases (70,000 °C/s) they are additional potent stimulators of RPE HSP activation [34,35,42,47,48]. Opposite USPL, SPL, and RPC, where therapeutic effectiveness and tissue damage run parallel and treatment becomes more effective as it becomes less safe/more damaging, by adjusting MPL parameters, safety and efficacy can be tailored and adjusted independently to be both maximally effective (developing an Arrhenius integral maximum of 1.0 for RPE HSP activation) and maximally safe (developing a very low Arrhenius integral for thermal cell death, such as <0.001, producing treatment reliably sublethal to the RPE) at the same time [35,47,48]. Thus, MPL has the unique potential to be both highly effective and clinically harmless at the same time (Figure 5). While the therapeutic range of USPL lasers is zero, and for SPL and RPC it is only 0.010 watt, the therapeutic range of extensively used MPL parameters can be 15 watts or more (Ojai Retinal Technologies, LLC, proprietary data) [33,35]. This is many times the maximum power output of current retinal laser systems, making it possible to safely and effectively treat the eyes of all patients using exactly the same laser parameters ("fixed" laser parameters) while strictly avoiding LIRD [34]. By eliminating per-eye laser exposure adjustments and thus possible surgeon misjudgments, safety is further maximized. At intracellular heating levels sublethal to the RPE (avg. temperature-rise of 6–8 °C, with pulse spikes of ~12–16 °C), the laser-induced temperature increases remain intracellular and there is no significant lateral, apical, or basal heat spread [48]. Unlike USPL, SPL, and RPC, because the therapeutic effects of all retinal laser modes arise in cells affected but not killed by treatment, MPL works directly rather than indirectly, eliciting these therapeutic effects only from the retina directly exposed to treatment (Figure 3). Thus, MPL has the capacity for maximizing the clinical response through *en masse* recruitment of large areas of dysfunction retina with confluent directly laser application [6,32]. This is forbidden for laser modes causing LIRD. Safe and effective direct treatment of the fovea, the most important locus for visually significant disease, is also made possible, as is ad lib retreatment, maximizing clinical utility—essential for the treatment of CPRs. Finally, the safety of appropriately designed MPL allows treatment of conditions that could not tolerate the trauma, inflammation, and loss of function inherent in LIRD; disorders such as inherited retinal degenerations and advanced AMD [6,34,54–64].

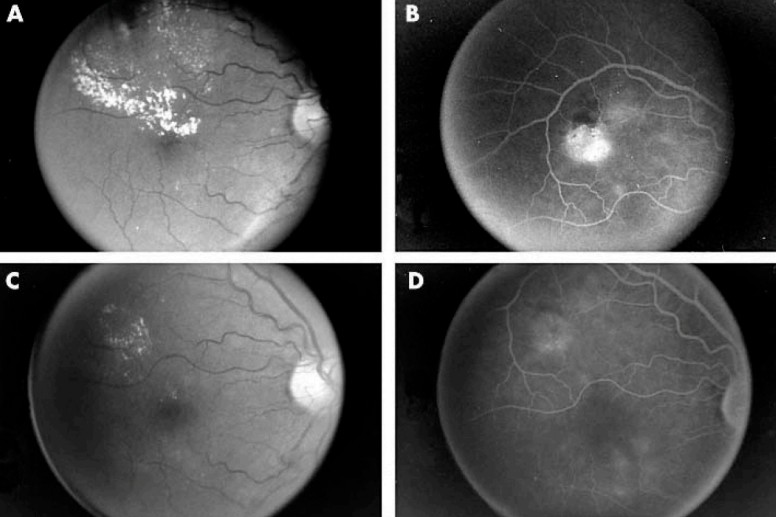

**Figure 5.** Red-free fundus photograph (**A**) and intravenous fundus fluorescein angiogram (**B**) of eye with clinically significant diabetic macular edema. (**C,D**) Same images after SDM laser treatment. Note resolution of diabetic macular edema without evidence of laser-induced retinal damage. From Luttrull JK, Much DC, Mainster MA. Subthreshold diode micropulse photocoagulation for the treatment of clinically significant diabetic macular oedema. Br. J. Ophthalmol. 2005;89(1):74–80. Doi:10.1136/bjo.2004.051540.

## 8. SDM and Modern Retinal Laser Therapy

Most early users of the first Micropulsed™ laser (Iridex, Mountain View, CA, USA) employed duty cycles of 15% and higher to increase the likelihood of causing LIRD with the 810 nm wavelength, thought to be essential to therapy. Because of this, low treatment densities, echoing those used with RPC, were retained, leading to the least effective retinal laser strategy of "low-intensity/low-density" treatment [28] (Figure 3). As a result, many dismissed MPL as clinically ineffective [2]. However, in 2000 "low-intensity/high density subthreshold

microsecond pulsed laser" (SDM) was developed, exploiting the unique facilities of the pulsed laser technology in a new way to perform retinal laser treatment entirely sublethal to the RPE and thus, for the first time completely absent of LIRD and still highly effective [4,6,64] (Figure 5). SDM epitomizes the principles of "modern retinal laser therapy" (MRT). MRT is defined as treatment that is reliably sublethal to the retina and thus clinically harmless, and optimized and made maximally clinically effective by confluent, high-density direct treatment of large areas of diseased retina [64].

## 9. The Clinical Implications of LIRD

The effectiveness of SDM without LIRD was instrumental in leading to our current understanding of the mechanism of retinal laser treatment as a physiologic "reset to default" phenomenon, triggered by thermal activation of RPE heat-shock proteins at temperatures that are sublethal to the directly exposed RPE cell, and in establishing the concepts and the first pillar of "modern retinal laser therapy" (MRT) [64]. RPE HSP activation, in turn, upregulates the endoplasmic reticulum unfolded protein response (ER UPR) to improve cell function, improve cell metabolism, inhibit apoptosis, and promote local and then systemic therapeutic immunomodulation [64]. Improved cell function necessarily results in normalized expression of, and response to, RPE-derived cytokines, interleukins, and other chemical mediators of retinal function and autoregulation. NIR lasers, acting only thermally, such as 810 nm employed in SDM, are anti-inflammatory [65–67]. An attribute of the reset mechanism is that it is agnostic to the cause of retinal dysfunction, which is different in every CPR, making SDM MRT a "non-specific trigger of disease-specific repair". Further, because activation of the ER UPR has no effect on normally functioning cells, SDM MRT is "pathoselective", affecting only dysfunctional cells. Lastly, MRT normalizes cell dysfunction in proportion to the pre-treatment degree of dysfunction. The greater the dysfunction, the greater the therapeutic effect [6,32,44,64]. The reset effects of normalizing cell and tissue function, independence from the cause of dysfunction, pathoselectivity, and proportionality with respect to the degree of pretreatment dysfunction, are arguably ideal attributes for any intervention.

As noted, the first pillar of MRT is functional normalization of tissue directly exposed to laser irradiation ("low-intensity" treatment) [6,17,32,64]. This is precluded by LIRD. The second conceptual pillar of MRT is equally problematic for LIRD: maximization and optimization of the clinical effects of treatment by confluent ("high-density") treatment of large areas of dysfunctional retina. It is obvious that such high-density treatment is forbidden for any laser causing LIRD as it would cause visual loss, if not blindness. The impact of even limited LIRD can be seen in the modest clinical effectiveness and limited applications for both ultra- and short-pulse lasers, and in the fact that such treatments do not improve retinal function by electrophysiology, but worsen it in proportion to the amount of treatment. [68] Clinically, the paucity of HSP activation by USPL and SPLs notably reduces their effectiveness for indications like DME, PDR, and other types of macular edema [27,38,44,69–75] (Figure 3). At the same time, the LIRD produced, while stimulating inflammatory debridement that can cause local drusen reduction, taxes the already comprised compensatory measures maintaining macular function in eyes with advanced dry AMD, accelerating disease progression and visual loss [26,55]. In contrast, SDM improves retinal (and optic nerve) function by electrophysiology and slows disease progression, thus lowering the risk of visual loss, especially in advanced diseases such as AMD [6,54–64] (Figures 5–7). This has been demonstrated in several studies showing a marked reduction in neovascular conversion in eyes with dry AMD managed with SDM VPT [55,56,59,60].

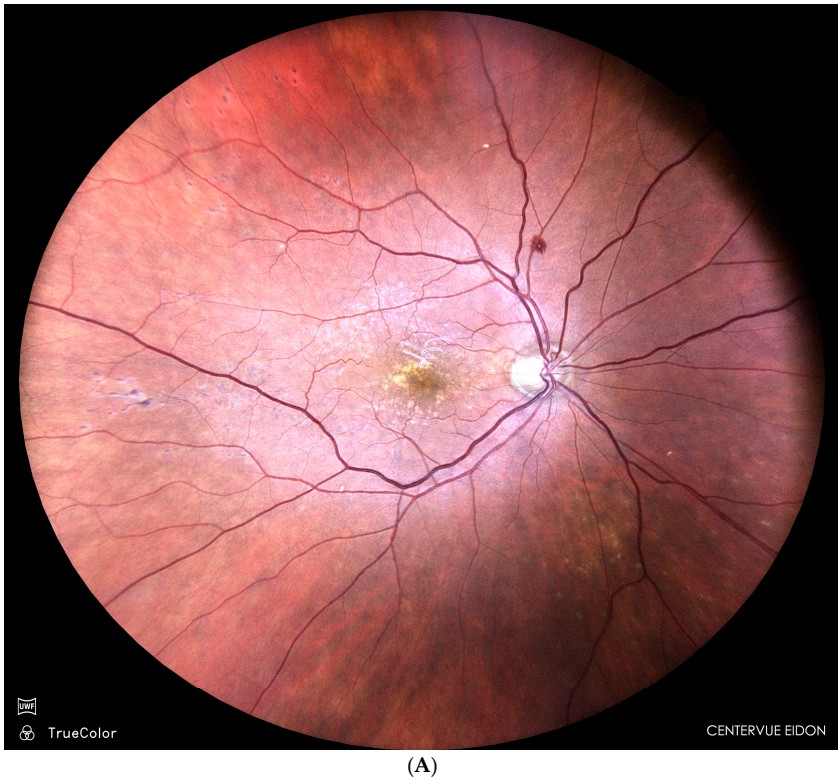

(**A**)

**Figure 6.** *Cont.*

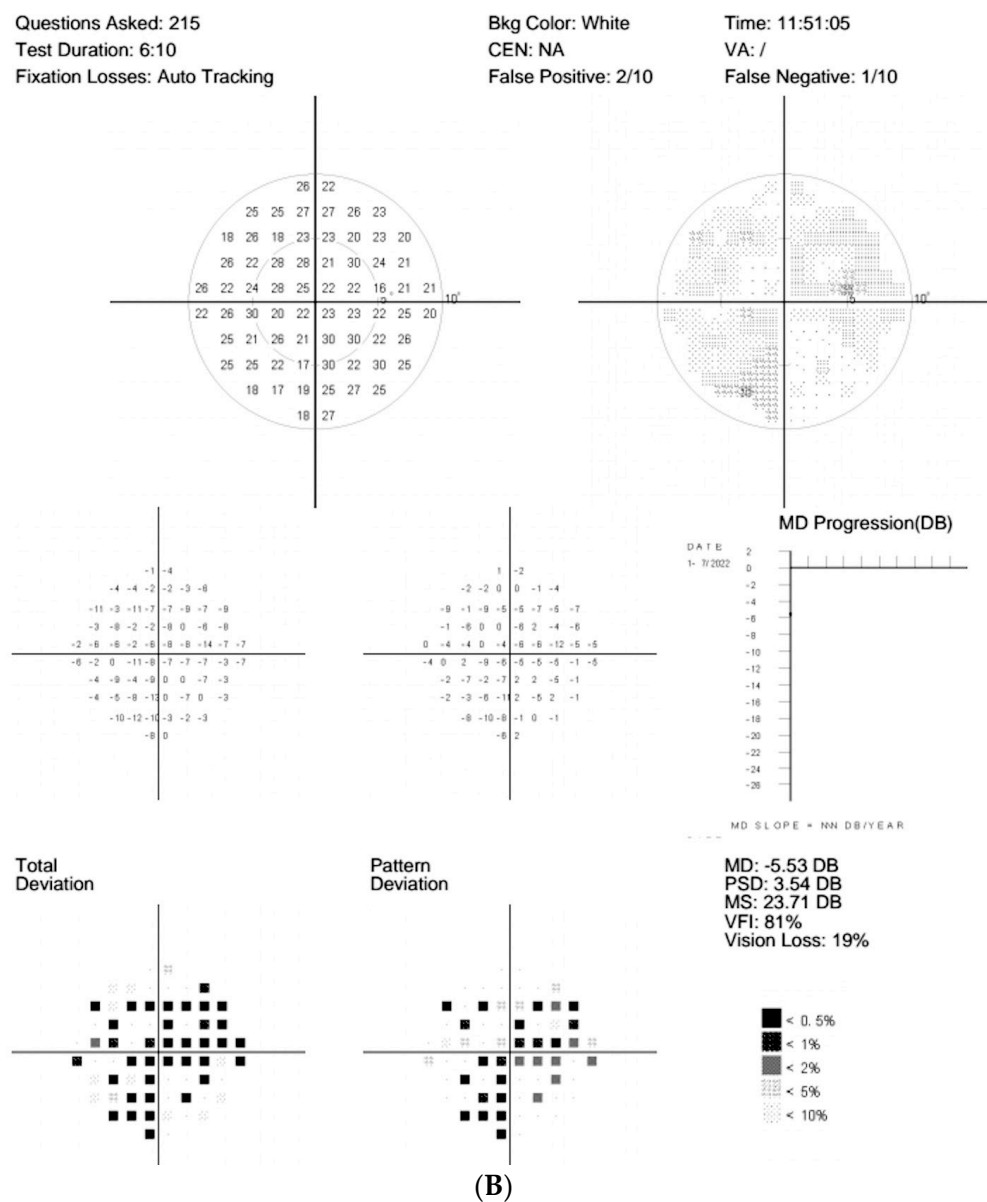

**Figure 6.** (**A**) Fundus photograph of eye of woman with intermediate dry age-related macular degeneration 1 year after pattern-scanning laser grid retinal photocoagulation. Note LIRD indicated by multiple small foci of LIRD in the superior and temporal extrafoveal macula. Visual acuity 20/25. (**B**) Automated perimetry demonstrating visual field loss as the result of the LIRD. Loss of visual function is an inherent result of LIRD. From: Luttrull JK. Modern Retinal Laser Therapy. Principles and Application. Kugler Publications, Amsterdam, The Netherlands August 2023 ISBN: 978-90-6299-298-0.

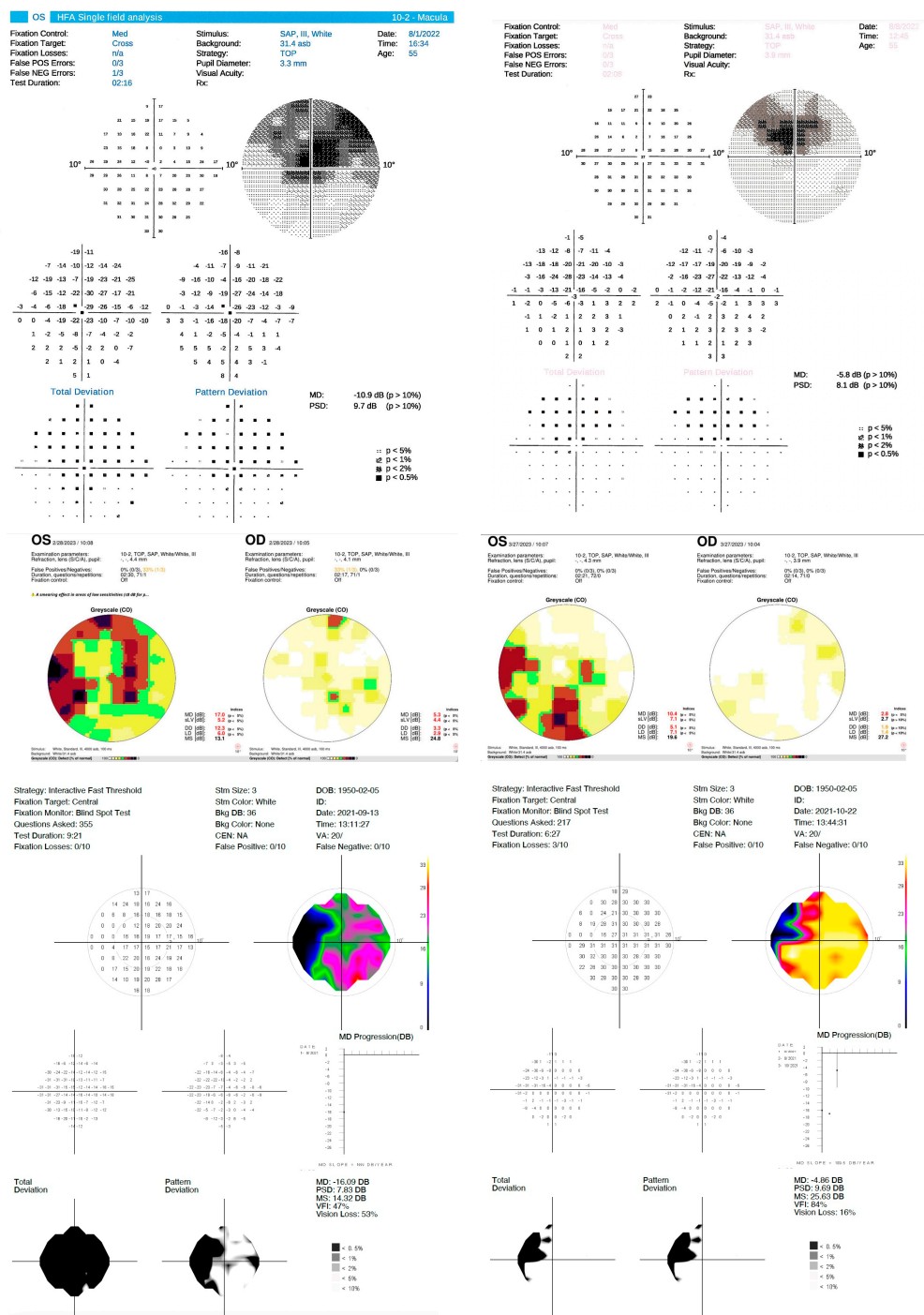

**Figure 7.** 10-2 Automated perimetry studies of eyes before panmacular SDM treatment. (**Top row**): left eye of male with superior scotoma due to post-branch retinal artery occlusion. Left, 3 months after occlusion. Right, 1 week following panmacular SDM. (**Middle row**): eye with open angle glaucoma before (left) and 3 weeks after (right) panmacular SDM. (**Bottom row**): both eyes of woman with optic atrophy and visual loss complicating multiple sclerosis, before (left) and 3 weeks after (right) panmacular SDM. Note that in each case, panmacular SDM resulted in significant recovery of visual function. SDM is the only therapeutic measure to improve visual fields in such settings. Avoidance of LIRD allows clinical exploitation of neuroprotective and neuroenhancing effects of retinal laser treatment to achieve such treatment benefits. From: Luttrull JK. Modern Retinal Laser Therapy. Principles and Application. Kugler Publications, Amsterdam, The Netherlands August 2023 ISBN: 978-90-6299-298-0.

## 10. The Ethical Implications of LIRD

Following Okham, it is axiomatic in medical ethics that given more than one treatment alternative for a given ailment with similar prospects for success, the one that is safest, simplest, least burdensome, and least invasive should be tried first [76]. Thus, one can think of the influence of LIRD on the outcome of retinal laser treatment in terms of a cost–benefit equation. In this case, the net effect of treatment would equal the therapeutic effects of treatment, which arise entirely from laser effects sublethal to the retina, minus the adverse treatment effects including loss of function, pain, and inflammation, and increased risks of visual loss which arise wholly from LIRD. In this equation, the only way that LIRD could improve outcomes would be if at least one of the effects of LIRD was sufficiently beneficial to outweigh the sum of the other negative effects. The fact that the list of unique effects of LIRD are uniformly detrimental makes this impossible. A number of studies have been reported comparing retina-sparing treatment (drug or SDM) to laser treatments causing LIRD, generally RPC, mostly for DME [71–75,77]. In these studies, mirroring the degree of adherence to the International Laser Society guidelines for subthreshold laser treatment, the visual results of retina-sparing treatments have been superior to lasers causing LIRD with clinical disease control, such as macular thickness reduction, and comparable to or superior to conventional RPC or drug therapy [33,71–75]. In the most recent randomized clinical trial comparing retina-sparing SDM MPL to conventional photocoagulation for DME, comparable results were found, but without adverse effects in the micropulse group [74]. This study confirmed, 18 years later, the first report on SDM for DME in 2005 [4]. Further, the authors noted that MPL results may have been improved if they had more fully incorporated MRT concepts such as foveal treatment [64,74]. Thus, as the mechanism of laser action anticipates, there is no evidence for a unique clinical benefit from LIRD compared to retina-sparing alternatives to justify the risks, adverse effects, and clinical limitations of LIRD.

## 11. Conclusions: The Impact of Elimination of LIRD on Clinical Disease Management

As the single source of all adverse laser treatment effects and risks, while making no unique therapeutic contributions to treatment, LIRD worsens visual outcomes compared to retina-sparing treatments, and severely limits laser treatment location, density, and repeatability, thus limiting effectiveness, application and treatment indications [64]. What is the clinical impact of laser treatment that does not cause LIRD?

As noted above, besides LIRD, all effects of retinal laser treatment are therapeutic. Clinical application of the reset effects of retinal laser treatment can now be applied to every CPR, including the traditional treatment indications of DR, DME, PDR, central serous retinopathy, and retinal vascular occlusions, and to new and important indications such as slowing the progression of IRDs, dry AMD, geographic atrophy, open angle glaucoma, and a host of other macular and even optic nerve dysfunctions [6,32–34,54–64] (Figures 6 and 7). Without LIRD, treatment effects can be optimized by high-density treatment and maintained by regular periodic treatment ("vision protection therapy") ad infinitum [6,32]. Without LIRD, direct treatment of the fovea, the most important locus in the eye, is possible, enhancing the ability to manage and prevent vision loss and vision-threatening diseases [34,45,48]. Finally, without LIRD, retinal laser treatment as MRT becomes the first safe and highly effective treatment modality suitable for preventive treatment of the most important causes of irreversible visual loss—the chronic progressive retinal neurodegenerations [64]. Early and preventive treatment allows for excellent vision to be preserved by avoiding vision loss, often difficult to restore, in the first place (Figure 8). We live in a pharmacologic world, and retinal laser treatment must often be considered along with drug therapy, usually intravitreal injection of a VEGF inhibitor. Studies have shown all forms of retinal laser treatment to reduce the need for drug therapy for indications such as complications of retinal vascular disease [38,65]. Because MRT allows earlier treatment, more extensive treatment, foveal treatment, and ad lib repeat treatment, MRT has the greatest potential to preclude the need for intravitreal drug injections altogether [65,71,73,75,77]. By slowing, stopping, or reversing disease progression, early and preventive treatment reduces severe disease that is more resistant to treatment and requires more invasive, expensive, and burdensome treatments. Thus, while LIRD was thought indispensable for over half a century, elimination of LIRD unlocks the full potentialities of retinal laser treatment to make it the most useful and broadly applicable sight-saving treatment modality in ophthalmology.

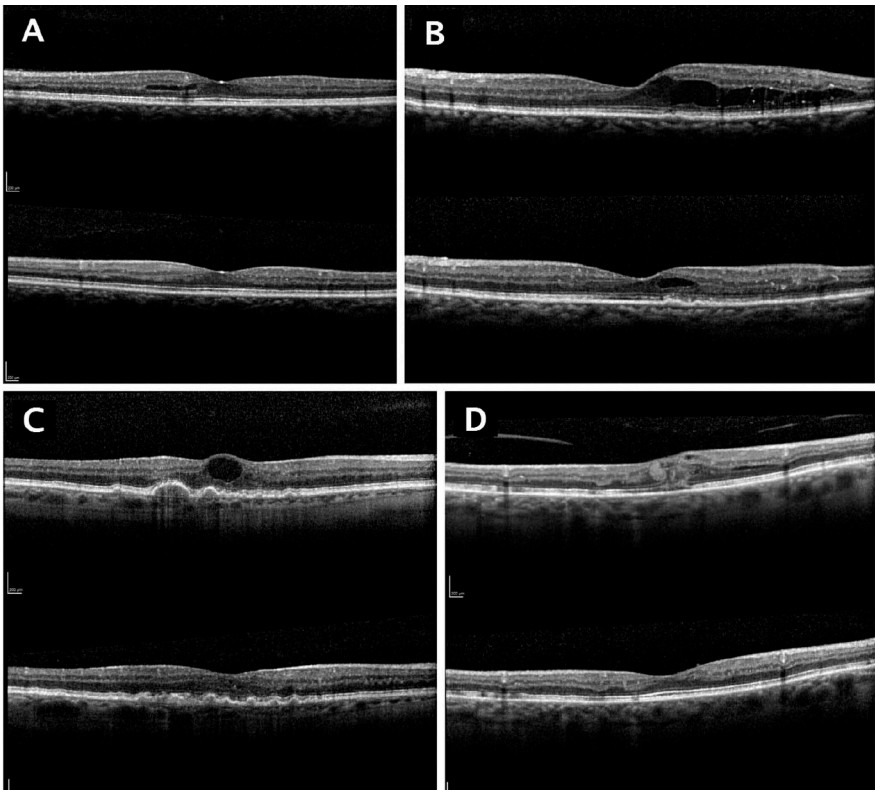

**Figure 8.** Spectral domain optical coherence tomographs (OCT) of eyes with center involving diabetic macular edema and pretreatment visual acuities of 20/40 or better. In each subfigure (**A–D)**, the top OCT image shows the pretreatment scan, while the bottom OCT image shows the post treatment scan. Note resolution of macular edema and the absence of retinal damage by OCT following transfoveal (panmacular) SDM laser treatment in each case. Despite excellent pretreatment visual acuities, visual acuities were significantly further improved following transfoveal laser treatment. Transfoveal treatment, treatment of eyes with excellent pretreatment visual acuities, and achieving significant visual acuity improvement in these eyes, were all made possible by the absence of LIRD with SDM. From Luttrull JK, Sinclair SD. Safety of transfoveal subthreshold diode micropulse laser for intra-foveal diabetic macular edema in eyes with good visual acuity. Retina, May 2014 Oct; 34 (10): 2010–2020.

**Funding:** This research received no external funding.

**Institutional Review Board Statement:** Not applicable.

**Informed Consent Statement:** Not applicable.

**Data Availability Statement:** Not applicable.

**Conflicts of Interest:** Ojai Retinal Technologies, LLC: management, equity; Retinal Protection Sciences, LLC: management, equity; Vision Protection Institutes, Inc: management, equity; Replenish, Inc: equity.

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
