# Peer review of "Lasers in Medicine: The Changing Role of Therapeutic Laser-Induced Retinal Damage—From de rigeuer to Nevermore"

_photonics, doi:10.3390/photonics10090999_

Round 1

Reviewer 1 Report

The manuscript entitled "photonics-2459049" dealing with the laser application has been reviewed. The paper has been nicely written but needs significant improvement. Please follow my comments. 1. Add some quantitative results to the abstract. 2. What is the main research question for this research work? 3. Figures 1 and 2 need scale bar. Please add it to the figure and briefly mention it in the text. 4. What is the future direction of this work? 5. Please proofread the text. 6. Short pulse laser in Page 8: Please add more information about this tool including the wavelength snd show if it can have the side effect on the body. 7. The paper doesn't have a conclusion. This is mandatory so please add a conclusion to your paper. 8. Lasers are used in different fields. For instance, medical and additive manufacturing processes. To highlight the application of laser in different fields read and add the following references in manufacturing and add a note in the introduction.. * "The effect of absorption ratio on meltpool features in laser-based powder bed fusion of IN718". * Multidisciplinary topology and material optimization approach for developing patient-specific limb orthosis using 3D printing * Numerical investigations on mechanical properties of bio-inspired 3D printed geometries using multi-jet fusion process

Needs some works.

Author Response

Comments and Suggestions for Authors

The manuscript entitled "photonics-2459049" dealing with the laser application has been reviewed. The paper has been nicely written but needs significant improvement. Please follow my comments. 1. Add some quantitative results to the abstract.

The following has been added to the abstract: “SDM allowed understanding of the mechanism of retinal laser as a physiologic reset effect triggered by heat-shock protein (HSP) activation upregulating the unfolded protein response and restoring proteostasis by increasing protein repair by 35% in dysfunctional cells via a thermally sensitive conformational change in the K10 step of HSP activation kinetics. Because HSP activation kinetics are catalytic even low levels (the “reset” threshold) of HSP activation result in a maximal treatment response. SDM and the study of HSP activation kinetics in the retina show that the therapeutic effects of retinal laser treatment can be fully realized without any degree of LIRD.  2. What is the main research question for this research work? “

As stated in the conclusion of the abstract: “Elimination of LIRD from retinal laser treatment has revolutionized the clinical potential of retinal laser treatment to broaden treatment indications and permit, for the first time, effective early and preventive treatment for all CPRs to reduce the most frequent causes of irreversible visual loss worldwide.”

  1. Figures 1 and 2 need scale bar. Please add it to the figure and briefly mention it in the text.

The following has been added to the legends of Figures 1 and 2: “Optic nerve identified by arrow. For scale, the horizontal diameter of the optic nerve is 1.5 mm.”

  1. What is the future direction of this work?

As described in the text, it is the expansion of the clinical usefulness and improvement in clinical results allowed by the discovery that LIRD is non-therapeutic, detrimental, and fully avoidable

  1. Please proofread the text.

Done (again)

  1. Short pulse laser in Page 8: Please add more information about this tool including the wavelength snd show if it can have the side effect on the body.

The follow has been added to the text: “SPL generally employs a 532nm wavelength.

The effect on the body is discussed, which is LIRD.”

  1. The paper doesn't have a conclusion. This is mandatory so please add a conclusion to your paper.

The word “Conclusion” has been added to the subtitle of the final section of the paper.

  1. Lasers are used in different fields. For instance, medical and additive manufacturing processes. To highlight the application of laser in different fields read and add the following references in manufacturing and add a note in the introduction.. * "The effect of absorption ratio on meltpool features in laser-based powder bed fusion of IN718". * Multidisciplinary topology and material optimization approach for developing patient-specific limb orthosis using 3D printing * Numerical investigations on mechanical properties of bio-inspired 3D printed geometries using multi-jet fusion process

The following has been added to the Introduction:This paper examines the transition from the belief that laser-induced retinal damage (LIRD) was an absolute prerequisite to therapeutic effectiveness, to the current understanding the LIRD is an unnecessary complication of treatment, how this was determined, and the implications of this transition from surgery to medicine.”

Reviewer 2 Report

1.       It’s confusing to read the whole text where “laser-induced retinal damage” is considered therapeutic. Based on my understanding, laser-induced retinal damage is the side effect of laser treatment.

2.       The text is too long and packed with too many details rather than summarization, which makes it hard to read. Please consider shorten the text.

3.       RPC is commonly referred as retinal progenitor cells, please consider rephrase panretinal photocoagulation as PRP.

Author Response

Comments and Suggestions for Authors

  1. It’s confusing to read the whole text where “laser-induced retinal damage” is considered therapeutic. Based on my understanding, laser-induced retinal damage is the side effect of laser treatment.

The reviewer is more sophisticated than most if he was aware of this prior to reading the paper. He is correct, and that is the point of the paper. The paper describes how LIRD was universally consider therapeutic for almost 6 decades, how it was determined that LIRD was actually an unnecessary complication of treatment, how most ophthalmologists are still unaware of this, and what it means.

  1. The text is too long and packed with too many details rather than summarization, which makes it hard to read. Please consider shorten the text.

Agreed. The text has been shorted by ~500  words

  1. RPC is commonly referred as retinal progenitor cells, please consider rephrase panretinal photocoagulation as PRP.

While word combinations are nearly infinite, acronyms are limited. Many acronyms are shared between different contexts. In the world of ophthalmology, RPC means “retinal photocoagulation”. We define it as such in the text and its use is appropriate for the context of the paper.

Reviewer 3 Report

Dr. Luttrull's review article, titled "Lasers in Medicine: The Changing Role of Therapeutic Laser-Induced Retinal Damage" is impressively well-written and offers deep insights. His clear and detailed writing demonstrates an admirable understanding of the topic. The article provides a significant contribution to the field, thoroughly exploring the subject of therapeutic laser-induced retinal damage.

I have several comments, primarily related to the article's formatting.

- Please select only five keywords.

- The "Introduction" section title is the only one in bold. Please ensure consistency by using bold formatting for all subsequent section titles as well. Alternatively, the author might consider adding numerical order to improve the readability and structure of the document (e.g., 1. Introduction, 2. Light for Cautery, and so on).

- On line 159, replace "CW lasers" with "continuous wave (CW) lasers".

- On line 303, please delete the phrase "(Figure 1 A-F from CDR 2012)".

- In Table 1, there are question marks in the first line of several columns. Could you please clarify what these represent?

- For images adapted from previously published papers, it is necessary to disclose permissions obtained from the original journal.

- Please ensure consistency in the formatting of references.

Author Response

Comments and Suggestions for Authors

Dr. Luttrull's review article, titled "Lasers in Medicine: The Changing Role of Therapeutic Laser-Induced Retinal Damage" is impressively well-written and offers deep insights. His clear and detailed writing demonstrates an admirable understanding of the topic. The article provides a significant contribution to the field, thoroughly exploring the subject of therapeutic laser-induced retinal damage.

I have several comments, primarily related to the article's formatting.

- Please select only five keywords.

Done

- The "Introduction" section title is the only one in bold. Please ensure consistency by using bold formatting for all subsequent section titles as well. Alternatively, the author might consider adding numerical order to improve the readability and structure of the document (e.g., 1. Introduction, 2. Light for Cautery, and so on).

Done. Thank you

- On line 159, replace "CW lasers" with "continuous wave (CW) lasers".

Done. Thank you

- On line 303, please delete the phrase "(Figure 1 A-F from CDR 2012)".

I apologize that I cannot find this in the text. This makes me wonder if the correct draft of the paper was submitted. This figure has become Figure 3

- In Table 1, there are question marks in the first line of several columns. Could you please clarify what these represent?

These were meant to improve understanding. For instance, LIRD? means: Does this cause LIRD? However, as it seems to have hindered more than helped, the question marks have been removed.

- For images adapted from previously published papers, it is necessary to disclose permissions obtained from the original journal.

 The these are either open access or out of copyright and the journals have been recognized.

- Please ensure consistency in the formatting of references.

Reviewed and done.

Reviewer 4 Report

The manuscript discusses the evolving role of therapeutic laser-induced retinal damage in medical lasers. It posits that laser-induced retinal damage is unnecessary and counterproductive to clinical results. Instead, it proposes that the advantages of retinal laser treatment can be wholly achieved without causing any damage. The removal of retinal damage from laser procedures has transformed its clinical potential, facilitating early and preventive interventions against irreversible visual loss.

General Observations:

Prior Reviews: I'd like to note that the manuscript appears to have undergone revisions. However, I do not have access to the first round of comments, and therefore, my feedback is based solely on the content presented in this revised version.

Overall Readability: The manuscript is well-composed and provides a comprehensive overview of the topic. The flow of information is logical, making it accessible and engaging for readers.

Specific Comments:

Role of Retina Photocoagulation: While the manuscript offers substantial insights into the topic, it does not address the crucial role of retina photocoagulation in reducing VEGF levels. This omission is particularly relevant since reducing VEGF directly impacts the risk of macular oedema in diabetic and RVO patients. I recommend that the author(s) incorporate this aspect to provide a more holistic understanding.

Comparative Analysis: Given the scope and relevance of the topic, it would be beneficial for readers if the author(s) could provide a comparative analysis of anti-VEGF agents and laser treatments. Specifically, this comparison should focus on their respective efficacies in addressing RVO and PDR neovascularisation as well as macular oedema associated with diabetes and RVO.

Recommendations:

Please elaborate on the role of retina photocoagulation in the context of VEGF reduction and its implications for macular oedema in diabetic and RVO patients.

Introduce a section or a subsection that offers a comparative study of anti-VEGF agents and laser treatments. Highlight the advantages, disadvantages, and the scenarios where one treatment might be preferable over the other.

Summary:

The manuscript, in its current form, provides valuable insights and adds to the existing literature on the topic. However, incorporating the aforementioned recommendations can make it even more robust and comprehensive. Once these changes are made, I believe the article will be a significant asset to the journal and its readership.

Author Response

Editor

Photonics

August 22, 2023

RE: photonics-2459049

Title: Lasers in medicine: the changing role of therapeutic laser-induced retinal damage. From de rigeuer to Nevermore

To the Editor,

Thank you for the opportunity to address the concerns of the final reviewer (no. 4). Point by point responses are provided below and noted in the revised manuscript with “track changes” for identification that we hope will make the paper suitable for publication in Photonics.

Specific Comments:

Role of Retina Photocoagulation: While the manuscript offers substantial insights into the topic, it does not address the crucial role of retina photocoagulation in reducing VEGF levels. This omission is particularly relevant since reducing VEGF directly impacts the risk of macular oedema in diabetic and RVO patients. I recommend that the author(s) incorporate this aspect to provide a more holistic understanding.

A major theme of the paper is that the therapeutic effects of all forms of laser treatment, whether photocoagulation or sublethal treatment, are the same. This includes reduction in VEGF and normalization of many other chemical factors of retinal and RPE origina as many of the references describe. To address the reviewer’s concern the following has been added to line 247: “These include reduction in vascular endothelial growth factor (VEGF) and increase in pigment epithelial derived factor. 32-38

Comparative Analysis: Given the scope and relevance of the topic, it would be beneficial for readers if the author(s) could provide a comparative analysis of anti-VEGF agents and laser treatments. Specifically, this comparison should focus on their respective efficacies in addressing RVO and PDR neovascularisation as well as macular oedema associated with diabetes and RVO.

I appreciate the reviewer’s interest in this topic, as it is certainly important. However, it is not the topic of the current paper and too vast in its own right to include in the current paper. To concisely address the reviewer’s interest ,the following has been added in lines 519-525: We live in a pharmacologic world, and retinal laser treatment must often be considered with respect to drug therapy, usually intravitreal injection of a VEGF inhibitor. Studies have shown all forms of retinal laser treatment to reduce the need for drug therapy for indications such as complications of retinal vascular disease. 38, 65 Because MRT allows earlier treatment, more extensive treatment, foveal treatment, and ad lib repeat treatment, MRT has the greatest potential to preclude the need for intravitreal drug injections altogether. 65, 71, 73, 75,77

Recommendations:

Please elaborate on the role of retina photocoagulation in the context of VEGF reduction and its implications for macular oedema in diabetic and RVO patients.

Please see above

Introduce a section or a subsection that offers a comparative study of anti-VEGF agents and laser treatments. Highlight the advantages, disadvantages, and the scenarios where one treatment might be preferable over the other.

Please see above. Greatly exceeds the scope of the current paper. I would refer the reviewer to my book on this topic for the most thorough discussion. (ref 65)

Summary:

The manuscript, in its current form, provides valuable insights and adds to the existing literature on the topic. However, incorporating the aforementioned recommendations can make it even more robust and comprehensive. Once these changes are made, I believe the article will be a significant asset to the journal and its readership.